# Nanoparticle anchoring targets immune agonists to tumors enabling anti-cancer immunity without systemic toxicity

Yuan Zhang[1,2], Na Li[1], Heikyung Suh[1] & Darrell J. Irvine[1,3,4,5,6]

Immunostimulatory agents such as agonistic anti-CD137 and interleukin (IL)−2 generate effective anti-tumor immunity but also elicit serious toxicities, hampering their clinical application. Here we show that combination therapy with anti-CD137 and an IL-2-Fc fusion achieves significant initial anti-tumor activity, but also lethal immunotoxicity deriving from stimulation of circulating leukocytes. To overcome this toxicity, we demonstrate that anchoring IL-2 and anti-CD137 on the surface of liposomes allows these immune agonists to rapidly accumulate in tumors while lowering systemic exposure. In multiple tumor models, immunoliposome delivery achieves anti-tumor activity equivalent to free IL-2/anti-CD137 but with the complete absence of systemic toxicity. Immunoliposomes stimulated tumor infiltration by cytotoxic lymphocytes, cytokine production, and granzyme expression, demonstrating equivalent immunostimulatory effects to the free drugs in the local tumor microenvironment. Thus, surface-anchored particle delivery may provide a general approach to exploit the potent stimulatory activity of immune agonists without debilitating systemic toxicities.

[1] Koch Institute for Integrative Cancer Research, MIT, Cambridge, MA, USA. [2] Department of Biomedical and Pharmaceutical Sciences, College of Pharmacy, University of Rhode Island, Kingston, RI, USA. [3] Department of Biological Engineering, MIT, Cambridge, MA, USA. [4] Department of Materials Science and Engineering, MIT, Cambridge, MA, USA. [5] Ragon Institute of MGH, MIT, and Harvard, Cambridge, MA, USA. [6] Howard Hughes Medical Institute, Chevy Chase, MD, USA. Correspondence and requests for materials should be addressed to Y.Z. (email: yuanzhang@uri.edu) or to D.J.I. (email: djirvine@mit.edu)

mmunostimulatory cytokines and antibodies are powerful anti-tumor therapeutics. However, systemic administration of immune agonists, including approved drugs such as interleukin-2 (IL-2) and interferon-α, are often accompanied by serious toxicities that limit dosing, and thereby efficacy[1,2]. Further, there is a strong immunological rationale for combining immune agonists to properly regulate immune responses—borne out by studies demonstrating synergistic enhancement of anti-tumor immunity with combination therapies[3–6]—but combinations of immune agonists can also exhibit further escalated toxicities[7]. Strategies are thus needed to enable immunostimulatory drugs to be used safely without compromising their anti-tumor activity.

Two highly synergistic immune agonists are IL-2 and agonistic antibodies or recombinant ligands for the co-stimulatory receptor CD137. Interleukin (IL)–2 stimulates the proliferation and effector function of cytotoxic T lymphocytes (CTLs) and natural killer (NK) cells, while CD137 (4–1BB) is a T-cell co-stimulatory receptor expressed by activated T-cells, NK cells, and a population of dendritic cells[8]. The CD137 ligand (CD137L) is a member of the tumor-necrosis factor (TNF) superfamily that binds CD137 to provide co-stimulatory signals for T-cell activation, and has been shown to have anti-tumor effects in a number of models when it binds to CD137 receptors to induce costimulation on T-cells[9]. Combined stimulation of IL-2 and CD137 receptors with IL-2 and anti-CD137 has been shown to enhance antigen-specific CD8+ T-cell responses[10]. We previously showed that combination treatment with IL-2 and agonistic anti-CD137 elicited potent anti-tumor immunity, but was also accompanied by severe systemic toxicity unless these agonists were confined to tumors by intratumoural injection of the cytokine and antibody covalently anchored to lipid nanoparticles, blocking their dissemination beyond the tumor and tumor-draining lymph nodes (TDLNs)[11]. However, this intratumoural treatment strategy would by definition be limited to accessible lesions, and unable to provide direct treatment of highly disseminated metastatic cancers.

In search of an approach to safely administer IL-2 and anti-CD137 to disseminated tumors that might further be generalizable to other combination therapies, here we test different systemic modalities for delivery of this combination treatment, and analyze root causes of the inflammatory toxicity of the combination therapy. We identify stimulation of circulating lymphocytes as a major source of the cytokine storm accompanying IL-2/anti-CD137 treatment. Together with our prior results indicating that these immune agonists have high efficacy and safety if confined to the tumor microenvironment[11], these results prompted us to test the utility of nanoparticle-IL-2/anti-CD137 formulations designed to rapidly accumulate in tumors while minimizing systemic exposure, through the enhanced permeation and retention (EPR) effect. To this end, we prepared stealth (PEGylated) liposomes bearing surface-conjugated IL-2 and anti-CD137. Treatment with combination liposomes leads to rapid accumulation of the immunostimulators in tumors but also accelerates clearance from the bloodstream compared to the free drugs. These combined features elicit potent activation of T-cell and NK cell responses in tumors equivalent to high doses of free IL-2/anti-CD137, while eliminating the cytokine storm and vascular leak syndrome (VLS) triggered by the free cytokine/antibody combination. This allows repetitive dosing of the immunoliposome forms leading to strong anti-tumor activity in the absence of systemic toxicity.

## Results

**Anti-CD137/IL-2-Fc combination is effective but toxic.** We focused on the poorly immunogenic, aggressive B16F10

melanoma model to evaluate combination treatments. Systemic administration of 20 µg IL-2 together with 100 µg anti-CD137 every 2 days for three doses had a modest impact on progression of established B16F10 tumors, and led to no improvement in survival (Fig. 1a, b). The efficacy of IL-2 can be enhanced by employing extended-pharmacokinetic (PK) forms of the cytokine[12,13], and thus we compared combination treatment by systemic administration of anti-CD137 together with wild-type murine IL-2 vs. an IL-2-Fc fusion with prolonged circulation half-life[12]. (The Fc domain of this fusion protein contained a D265A mutation to ablate Fcγ receptor binding.) Administration of anti-CD137 and IL-2-Fc to tumor-bearing mice under the same treatment schedule initially controlled tumor progression (Fig. 1a), but this enhanced therapeutic response was accompanied by severe systemic toxicity, with rapid weight loss initiated following the first injection that accumulated with each dose, leading to death following the third injection on day 14 (Fig. 1b, c). Increasing the time interval between doses of anti-CD137 and IL-2-Fc failed to alleviate the severe toxicity of the combination treatment, while administration of lower doses of anti-CD137/IL-2-Fc lowered the degree of toxicity but also reduced the efficacy in slowing tumor progression (Fig. 1a–c, Supplementary Fig. 1a–c), suggesting that anti-tumor efficacy and toxicity were linked in this systemic therapy.

**Toxicity derived from circulating lymphocytes and VLS.** Systemic cytokine release is a common side effect of many immune agonists in animal models and also in humans[14], and we found that anti-CD137/IL-2-Fc combination therapy triggered a dose-dependent systemic cytokine storm, with significant elevations of several inflammatory cytokines and chemokines (Fig. 1d). T-cells and NK cells have been implicated previously in side effects of IL-2[15] and anti-CD137[16] monotherapies, and thus we assessed the involvement of these cells in the toxicity of the combination treatment. Administration of anti-CD137 and IL-2-Fc to NOD/scid/gamma chain mice that lack mature T-cells, B cells, and NK cells, elicited negligible levels of systemic inflammatory cytokines/chemokines (Supplementary Fig. 2a). To determine whether circulating lymphocytes play an important role, tumor-bearing mice were treated with IL-2-Fc/anti-CD137 in the presence of FTY720, a sphingosine-1-phosphate analog that depletes both T-cells and NK cells from the peripheral blood by blocking their egress from lymph nodes[17,18]. As expected, FTY720 treatment dramatically lowered the number of CD4+ and CD8+ T-cells in blood, and partially reduced NK cell levels during treatment (Supplementary Fig. 2b). FTY720-treated animals showed greatly lowered systemic cytokine release in response to the combination therapy, suggesting that sustained stimulation of circulating lymphocytes plays an important role in the cytokine storm elicited by the combination therapy (Fig. 1e).

To determine the relative roles of T-cells and NK cells and evaluate cytokine/chemokine levels over time during anti-CD137/IL-2-Fc therapy, we depleted CD8+ T-cells or NK cells from C57Bl/6 mice bearing B16F10 tumors, then began systemic anti-CD137/IL2-Fc therapy on day 8 after tumor inoculation (Supplementary Fig. 3a). In animals receiving an isotype control in lieu of depleting antibodies, the combination therapy triggered sustained inflammatory cytokine/chemokine levels in serum that either increased or were sustained from day 10 to day 13 post tumor inoculation, while depletion of either CD8+ T-cells or NK cells during treatment in wild-type mice greatly reduced the cytokine storm elicited by the combination therapy at these time points (Supplementary Fig. 3b). However, depletion of either NK or CD8 cells during anti-CD137/IL2-Fc therapy was still accompanied by gross weight loss in treated animals

(Supplementary Fig. 3c), suggesting additional mechanisms of toxicity beyond the cytokine storm produced by these lymphocytes. IL-2 has previously be reported to elicit vascular damage through direct stimulation of endothelial cells or indirectly through granulocytes[15,19–21]. Indeed, IL-2-Fc/anti-CD137 combination therapy induced edema characteristic of VLS in the lung

1 day after the second i.v. injection (Fig. 1f). One day after three injections, IL-2-Fc/anti-CD137 combination therapy resulted in increased interstitial pulmonary infiltrates (Fig. 1g). Thus, IL-2-Fc/anti-CD137 combination therapy induces both a systemic cytokine release dependent on circulating T and NK cells and VLS.

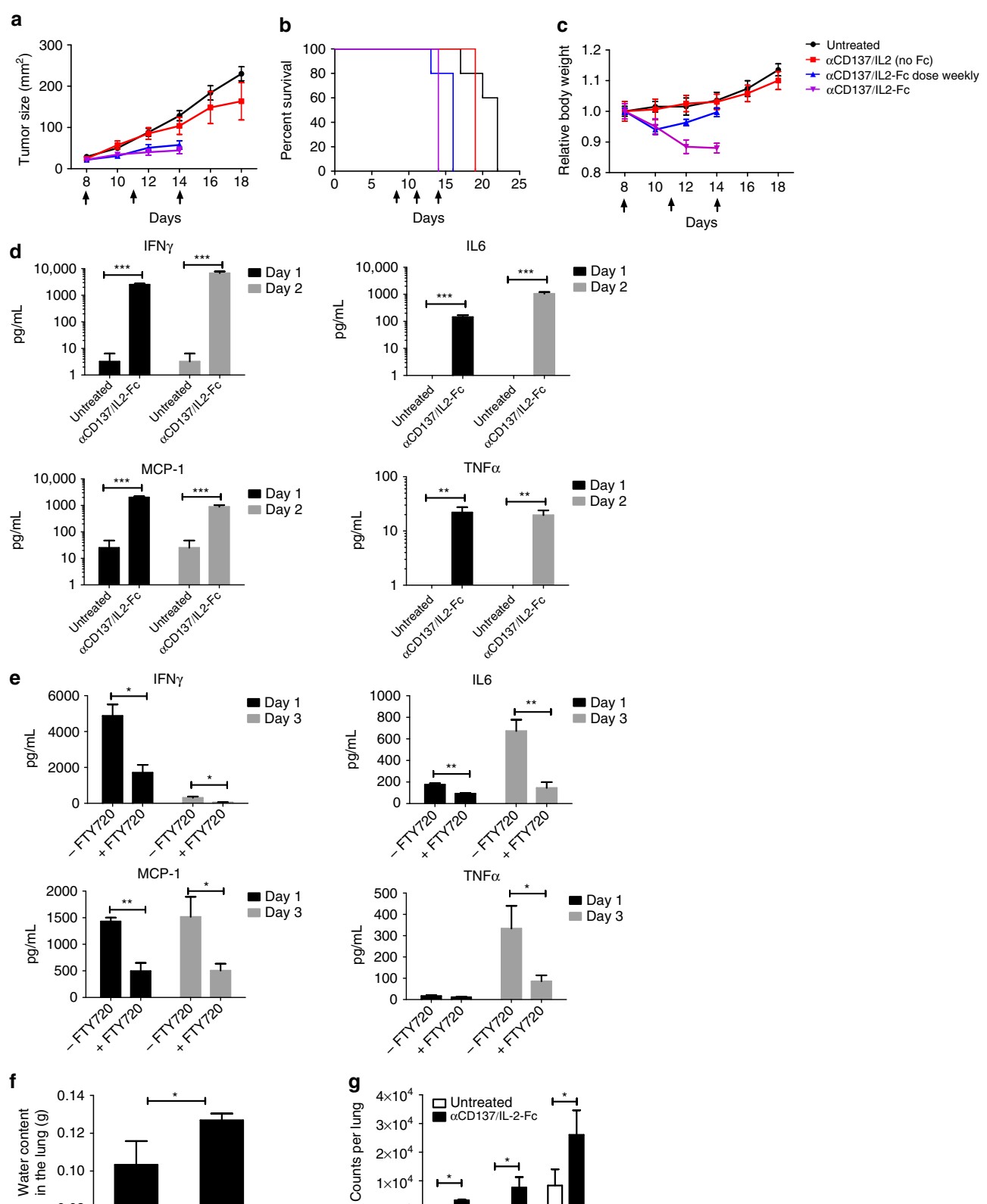

**Synthesis of anchored antibody immunoliposomes.** Given the toxicity of the systemic free drug combination, we considered strategies to concentrate IL-2 and anti-CD137 in tumors with less systemic exposure. We hypothesized that conjugation of these immune agonists to nanoparticles could promote their redistribution to tumors through the enhanced permeability and retention effect, while lowering their persistence in the systemic circulation relative to the free drugs. To test this idea, we coupled IL-2-Fc and $F(Ab')_2$ fragments of anti-CD137 to PEGylated liposomes via hinge region thiols of the two molecules. PEGylated liposomes conjugated with isotype-matched IgG antibody were used as controls. The resulting immunoliposomes were unilamellar vesicles with sizes of $80 \pm 22$ nm and zeta potentials of $-25$ mV as measured by cryogenic transmission electron microscopy (cryo-TEM) and dynamic light scattering (DLS) (Fig. 2a). From the conjugation efficiency, per 1 mg lipids, there were ~123 µg anti-CD137 or ~99 µg IL2-Fc conjugated on the liposome surfaces. The small size allows better tumor penetration via the EPR effect, while PEGylation reduces serum protein opsonization to minimize phagocytosis by reticuloendothelial system and premature particle clearance before reaching the target site in tumors[22]. Both flow cytometry and confocal imaging showed that fluorescently labeled liposome-conjugated anti-CD137 (Lipo-αCD137) and liposome-conjugated IL-2-Fc (Lipo-IL-2-Fc) bound at high levels to activated T-cells compared to control IgG-conjugated liposomes (Lipo-IgG) after a 3 h incubation in vitro (Supplementary Fig. 4a, b). Lipo-IL-2-Fc and Lipo-αCD137 stimulated T-cells similarly to free IL-2-Fc and anti-CD137, as measured by the $CD8^+$ T-cell counts and IFN-γ secretion, respectively, in vitro (Fig. 2b, c). These data suggest that the conjugation of IL-2-Fc and anti-CD137 to liposome surface via maleimide coupling did not compromise their biological activity and binding affinity to their corresponding receptors.

We further measured the liposome stability and breakdown in vitro. IRDdye-labeled anti-CD137 or IL-2-Fc were conjugated to liposomes carrying encapsulated Alexa Fluor-labeled ovalbumin as a marker of intact vesicles, and ratios of anti-CD137 or IL-2-Fc remaining bound to liposomes in serum-containing buffer were measured over time. Normalized to the initial time point, both IL-2-Fc and anti-CD137 levels remained approximately constant over two days, suggesting the immune agonists were stably conjugated on the liposome surface over time (Supplementary Fig. 5).

**Liposome delivery alters therapy PK and biodistribution.** We first compared the in vivo pharmacokinetics and biodistributions of IL-2-Fc and anti-CD137 carried by liposomes vs. the free molecules in mice bearing subcutaneous tumors. Liposomal anti-CD137 exhibited a total systemic exposure (area under the curve (AUC)) ~60% of free anti-CD137 in the blood, while Lipo-IL-2-Fc exhibited significantly faster clearance than IL-2-Fc, with an AUC approximately half of the free cytokine (Fig. 2d, e, Table 1). At 24 h, the amount of circulating Lipo-IL-2-Fc in the blood was much lower than the amount of IL-2-Fc (Fig. 2e). We also observed that although soluble and liposomal IL-2-Fc and anti-CD137 exhibited similar binding to circulating NK cells at 24 h, substantially reduced levels of Lipo-IL-2-Fc associated with T-cells in the blood at 24 h post injection compared to IL-2-Fc (Fig. 2j). Similar differences in binding of soluble and liposome-bound IL-2-Fc were observed when anti-CD137 and IL-2-Fc were administered simultaneously. Examining their biodistributions more broadly, Lipo-IL-2-Fc exhibited higher accumulation in the spleen and lymph nodes than IL-2-Fc, and Lipo-αCD137 had higher kidney accumulation than anti-CD137, while accumulation in other organs was largely similar for liposomal vs. free proteins (Fig. 2f, g). Thus, the liposomal agonists cleared from the blood more quickly than their free counterparts, with Lipo-IL-2-Fc particularly exhibiting more rapid clearance from the blood than soluble IL-2-Fc.

Since the strong absorption across the visible spectrum for melanin[23] made bulk fluorescence-based measurement of tumor uptake of the labeled IL-2 and anti-CD137 unreliable, an unpigmented B16F10 Trp2 knockout (KO) cell line[3] (B16F10-Trp2KO) that does not produce melanin was used as a substitute for B16F10 in tumor accumulation experiments for the quantitation of the % injected dose that localized to the tumor. In the B16F10-Trp2KO xenografts, after bolus i.v. injection, liposomal anti-CD137 showed a approximately fivefold increase of tumor accumulation compared to free anti-CD137 at the 4 and 24 h time points (Fig. 2h). Liposomal IL-2-Fc showed a ~50% increase in tumor accumulation comparing to that of free IL-2-Fc at 4 h, while accumulations of liposomal and free IL-2-Fc were comparable at 24 h (Fig. 2i). We further examined the tumor permeation of the soluble or liposomal agonists microscopically in tissue sections. Animals bearing subcutaneous B16F10 flank tumors were administered i.v. injections of fluorescently labeled soluble proteins or protein-anchored liposomes, followed by killing and tissue sectioning for confocal imaging of tumor tissues. We observed striking differences in their tumor accumulation at 1 h post administration. Liposomal immune agonists displayed substantially greater accumulation in tumors compared to their soluble forms (Fig. 3a, b). Quantification of total anti-CD137/IL-2-Fc signal in tumor sections showed at 4.5-fold increase in uptake of the combined therapeutics in the liposomal form compared to the free drugs (Fig. 3a, $p < 0.001$). Thus, liposomes showed more rapid clearance from the blood accompanied by enhanced tumor accumulation at early times—indicating lower systemic exposure but retention of intratumoural delivery of these agonists.

**Immunoliposomes elicit anti-tumor efficacy without toxicity.** We next evaluated the relative anti-tumor efficacy and safety of

**Fig. 1** Anti-CD137 and IL-2-Fc combination immunotherapy effectively arrests tumor growth, but elicits lethal in vivo toxicity. Groups of C57BL/6 mice ($n = 5$ mice/group) were subcutaneously inoculated with $5 \times 10^5$ B16F10 cells, and received i.v. injections of αCD137 and IL-2 or αCD137 and IL-2-Fc on days 8, 11, and 14, at a dose of 100 µg anti-CD137 and 20 µg IL-2 or 60 µg IL-2-Fc (equivalent mole dose to 20 µg IL-2). A third group received αCD137 and IL-2-Fc only on days 8 and 15. Shown are tumor sizes (**a**), survival (**b**), and body weight (normalized to day 8) (**c**). **d** B16F10 tumor-bearing mice received i.v. injections of αCD137 + IL-2-Fc. One and 2 days later, sera from peripheral blood were collected, and inflammatory cytokine and chemokine levels in sera were measured by luminex cytokine assays. **$p < 0.005$, ***$p < 0.0005$. **e** Groups of C57BL/6 mice ($n = 5$ mice/group) were subcutaneously inoculated with $5 \times 10^5$ B16F10 cells and were given daily i.p. injection of FTY720 starting day 7 to deplete circulating lymphocytes. Mice received i.v. injections of αCD137 and IL-2-Fc on days 8. One day and 3 days later, sera from peripheral blood were collected and inflammatory cytokine/chemokine levels in sera were measured by luminex cytokine assays. *$p < 0.05$, ** $p < 0.005$. **f, g** Mice bearing subcutaneous melanoma received i.v. injections of αCD137 and IL-2-Fc on days 8 and 10. One day later, lungs were collected and freshly weighed before lyophilization. The lyophilized lungs were weighed, and the water content in the lung was calculated (**f**). *$p < 0.05$. Lymphocyte infiltrates in the lung were analyzed using flow cytometry 1 day after the last injection (**g**). *$p < 0.05$. All measurements shown are mean ± s.e.m.

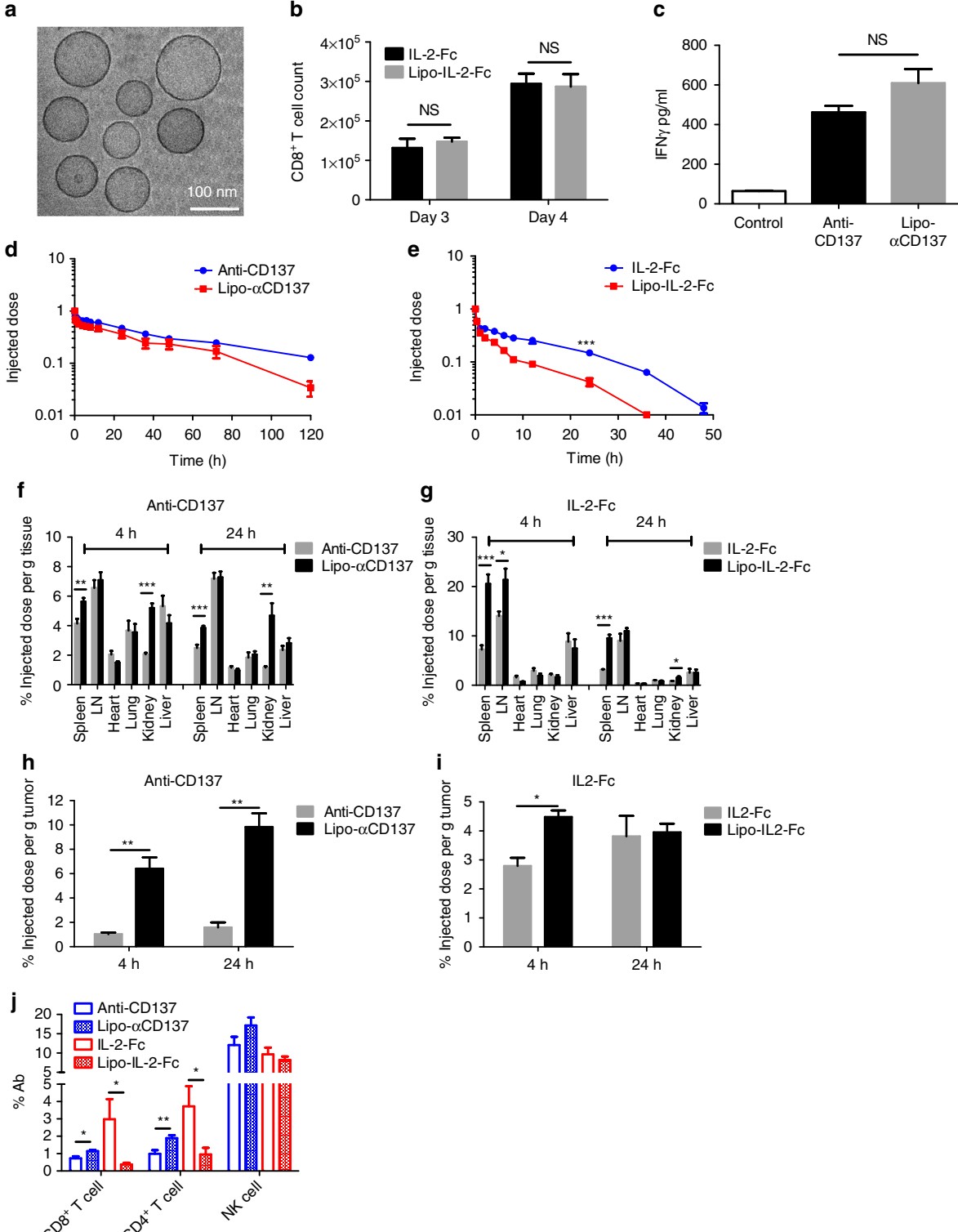

**Fig. 2** Liposome-conjugated anti-CD137 and IL-2-Fc retain bioactivity with lower systemic exposure than the free drugs. **a** Representative cryo-TEM image of IL-2-Fc-liposome (anti-CD137 liposomes were qualitatively identical). **b** Polyclonal T-cells from C57Bl/6 mice were activated for 2 days with anti-CD3/CD28 beads, then soluble or liposomal IL-2-Fc (10 ng/mL of protein) were added, and CD8[+] and CD4[+] T-cell counts were determined on days 3 and 4. **c** Activated T-cells were incubated with soluble anti-CD137 or Lipo-αCD137 (final αCD137 concentration: 10 μg/mL) for 6 h, and the secreted IFN-γ in the cell culture supernatant was analyzed by ELISA. **d–g** Groups of C57Bl/6 mice were injected s.c. with 5 × 10[5] B16F10 tumor cells, and then received i.v. injections of fluorescently labeled soluble IL-2-Fc, soluble anti-CD137, Lipo-αCD137, or Lipo-IL-2-Fc proteins (100 μg αCD137, 60 μg IL-2-Fc) on day 10. Pharmacokinetics of the labeled proteins were followed over time in the blood (**d**, **e**) and protein levels in the organs were compared at 4 and 24 h later (**f**, **g**). *$p < 0.05$, **$p < 0.005$, ***$p < 0.0005$. **h**, **i** The anti-CD137 (**h**) and IL-2-Fc (**i**) accumulation in unpigmented B16F10-Trp2KO xenograft tumors were determined at 4 and 24 h after bolus i.v. injection. *$p < 0.01$, **$p < 0.005$. **j** Twenty-four hours after i.v. injections of fluorescently labeled free or liposomal αCD137 and IL-2-Fc, lymphocytes from peripheral blood were collected, and their association with fluorescent protein drugs were analyzed by flow cytometry. *$p < 0.05$, **$p < 0.01$. All measurements shown are mean ± s.e.m. NS not significant

### Table 1 Pharmacokinetic parameters of the PK study

|  | A | $\alpha$ (per h) | B | $\beta$ (per h) | $t_{1/2,\alpha}$ (h) | $t_{1/2,\beta}$ (h) | AUV |
|---|---|---|---|---|---|---|---|
| IL2-Fc | $0.57 \pm 0.08$ | $3.66 \pm 1.00$ | $0.43 \pm 0.08$ | $0.05 \pm 0.01$ | $0.20 \pm 0.03$ | $13.22 \pm 2.95$ | $8.27 \pm 2.40$ |
| Lipo-IL2-Fc | $0.72 \pm 0.04$ | $2.19 \pm 0.33$ | $0.28 \pm 0.04$ | $0.09 \pm 0.01$ | $0.32 \pm 0.05$ | $7.52 \pm 0.83$ | $3.37 \pm 0.26$ |
| $\alpha$CD137 | $0.31 \pm 0.15$ | $3.70 \pm 2.59$ | $0.69 \pm 0.15$ | $0.02 \pm 0.01$ | $0.31 \pm 0.25$ | $44.12 \pm 10.84$ | $41.95 \pm 2.68$ |
| Lipo-$\alpha$CD137 | $0.41 \pm 0.13$ | $4.79 \pm 2.36$ | $0.59 \pm 0.13$ | $0.02 \pm 0.01$ | $0.18 \pm 0.10$ | $32.07 \pm 8.51$ | $28.09 \pm 12.53$ |

$t1/2,\alpha$ fast clearance half-life, $t1/2,\beta$ slow clearance half-life, AUC area under the curve

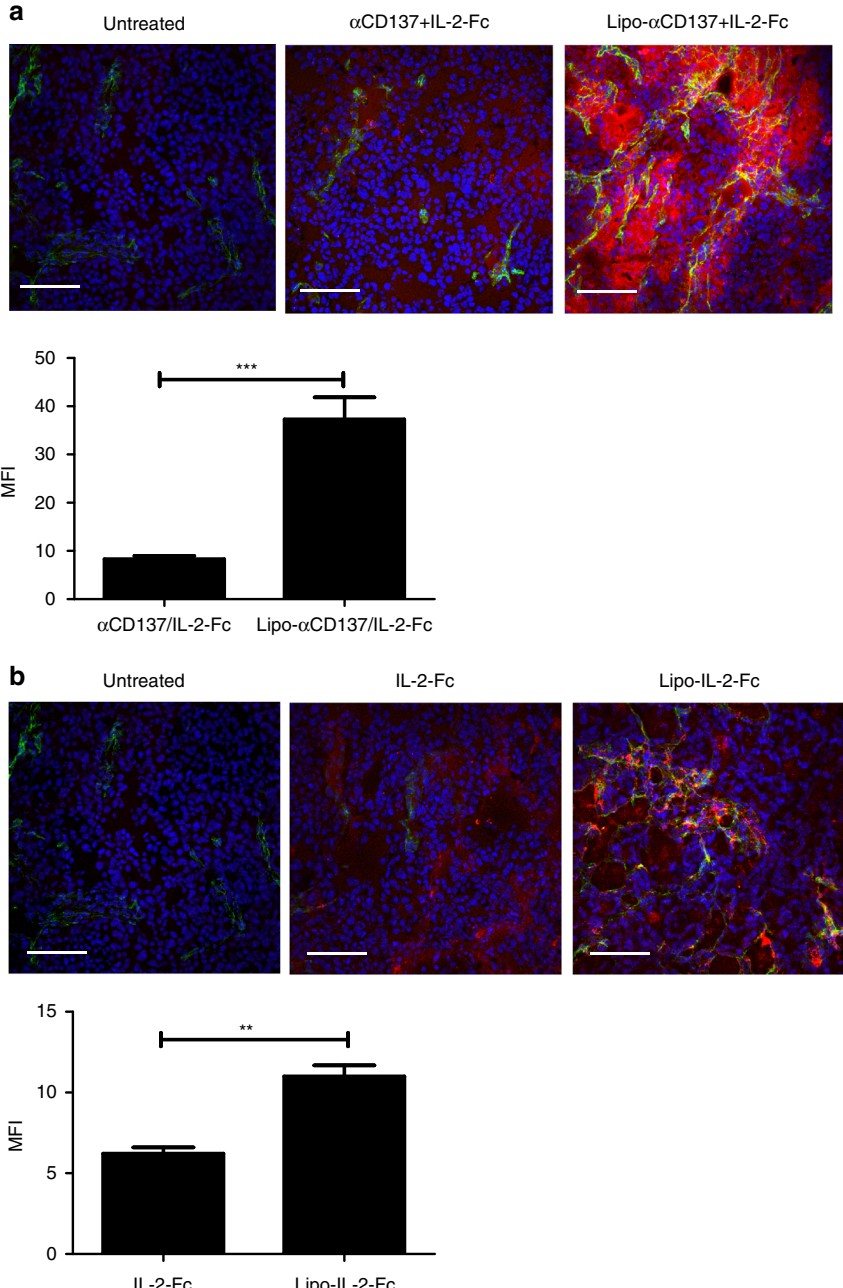

**Fig. 3** Liposome-anchored anti-CD137 and IL-2-Fc rapidly accumulate in tumors. **a**, **b** Groups of C57Bl/6 mice ($n = 3$/group) were inoculated with $5 \times 10^5$ B16F10 tumor cells on day 0, and received i.v. injections of Alexa-568-labeled $\alpha$CD137 and IL-2-Fc, Lipo-$\alpha$CD137 + Lipo-IL2-Fc, IL-2-Fc alone, or Lipo-IL-2-Fc alone on day 10. One hour later, tumors were collected, and frozen sections were stained for CD31 and DAPI and imaged by confocal microscopy. Background-corrected fluorescence intensities of Alexa 568 in tumors were quantified by ImageJ. Scale bar, 100 μm. Blue: nuclei; green: CD31 staining; red: Alexa 568 labeled anti-CD137 or IL-2-Fc. **$p < 0.001$, ***$p < 0.0001$. Measurements shown are mean ± s.e.m.

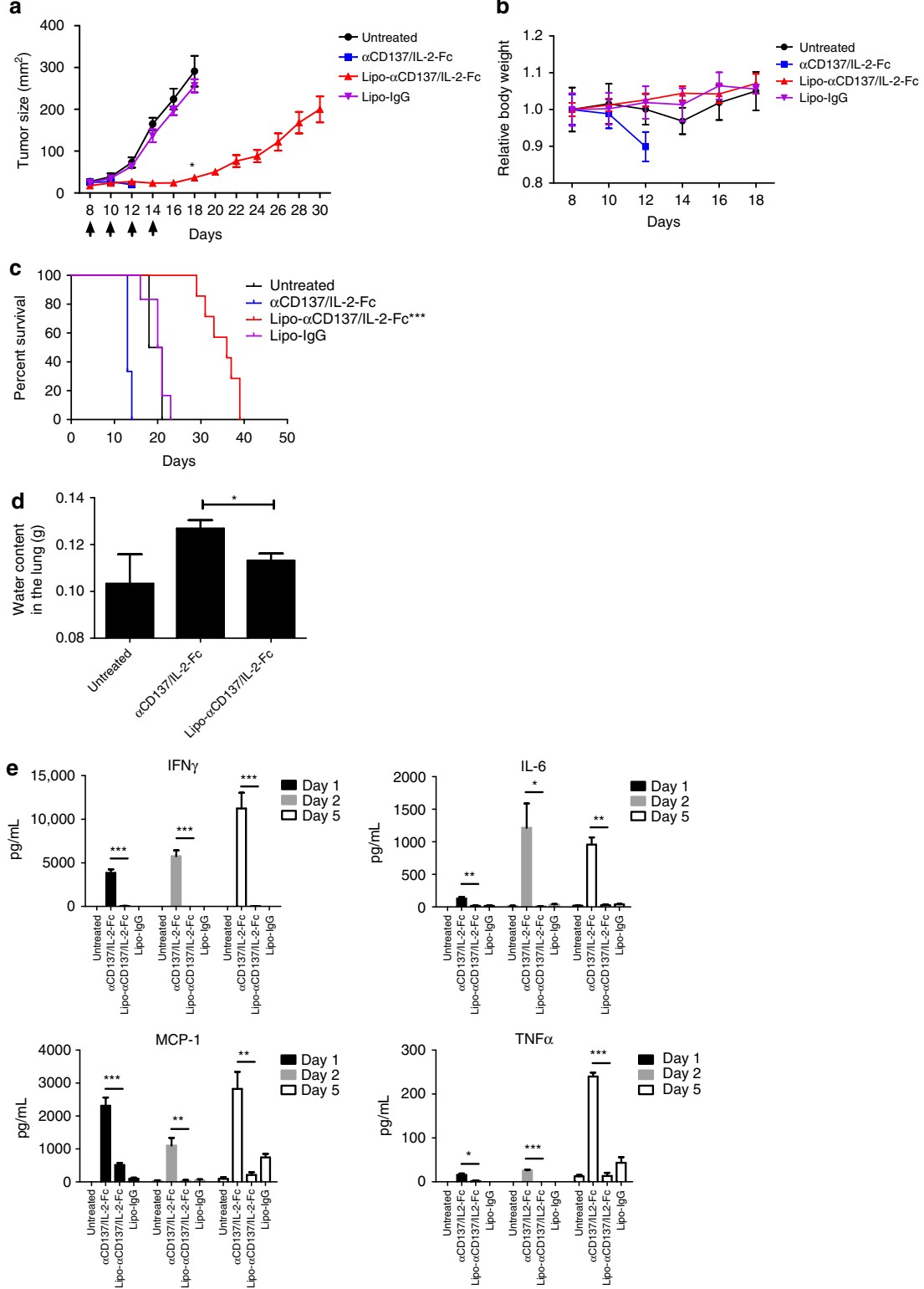

**Fig. 4** Immunoliposome IL-2-Fc/αCD137 therapy inhibits melanoma tumor growth and prolongs survival without toxicity. **a**–**c** Groups of C57Bl/6 mice ($n = 6$–7 per group) were inoculated with $5 \times 10^5$ B16F10 tumor cells s.c. on day 0, then received i.v. injections of αCD137 + IL-2-Fc, Lipo-αCD137 + Lipo-IL-2-Fc, or Lipo-IgG on days 8, 10, 12, and 14 after tumor cell inoculation (100 µg per dose αCD137 and 60 µg per dose IL-2-Fc). Shown are mean tumor sizes (**a**, *$p < 0.0005$ Lipo-CD137/IL-2-Fc vs. untreated and Lipo-IgG), relative body weight (**b**, normalized to day 8), and overall survival **c** vs. time. ***$p < 0.001$ Lipo-CD137/IL-2-Fc vs. untreated, αCD137/IL-2-Fc and Lipo-IgG. **d** Groups of C57Bl/6 mice ($n = 4$/group) were inoculated with $5 \times 10^5$ B16F10 tumor cells s.c. on day 0, then were treated on days 8 and 10 as in **a**. On day 11, animals were killed and water content in the ling was measured. *$p < 0.05$. **e** Groups of C57Bl/6 mice ($n = 5$/group) were inoculated with $5 \times 10^5$ B16F10 tumor cells s.c. on day 0, then were treated on days 8, 10, and 12 as in **a**. On days 9, 10, and 13, sera were collected from peripheral blood and analyzed for levels of cytokines and chemokines by luminex assays. *$p < 0.05$, **$p < 0.005$, ***$p < 0.0001$. Shown are mean ± s.e.m.

immunoliposomes compared to the free IL-2-Fc/αCD137 combination therapy. B16F10 tumor-bearing mice were administered free or liposomal anti-CD137 and IL-2-Fc combinations four times every other day. As shown in Fig. 4a, Lipo-αCD137/Lipo-IL2-Fc combination therapy significantly delayed tumor outgrowth, while isotype control liposomes (Lipo-IgG) had no impact on tumor progression. Importantly, this halt in tumor growth was achieved without evidence of toxicity from animal body weights, in contrast to the free agonists, which induced steady weight decline to animal mortality as before (Fig. 4b). Correlating with these distinct outcomes, immunoliposome treatment led to significantly enhanced survival compared to untreated tumors, while soluble anti-CD137 and IL-2-Fc treatment led to reduced survival due to immunotoxicity (Fig. 4c). One day after the second injection, mice treated with αCD137/IL-2-Fc had a significant elevation of serum levels of cholesterol and the liver enzymes aspartate transaminase (AST) and alanine transaminase (ALT), as well as elevated serum creatinine kinase, suggesting liver and kidney damage (Supplementary Fig. 6a–d). By contrast, Lipo-αCD137 + Lipo-IL-2-Fc elicited no changes in any of these markers above baseline. Further, 1 day after two injections, mice treated with Lipo-αCD137 + Lipo-IL-2-Fc showed alleviated VLS compared to the anti-CD137 + IL-2-Fc group, as indicated by water content in the lungs (Fig. 4d). Immunoliposome therapy also showed no evidence of systemic cytokine storm induction, as evidenced by levels of proinflammatory cytokines in serum (Fig. 4e). To assess the generality of these findings, we carried out similar studies in balb/c mice bearing A20 B cell lymphoma tumors. Tumor growth was effectively suppressed after the treatment with either αCD137/IL-2-Fc or Lipo-αCD137/Lipo-IL-2-Fc (Supplementary Fig. 7a). However, soluble αCD137/IL-2-Fc therapy led to steadily accumulating weight loss and elevated levels of inflammatory cytokines in the circulation, while the liposomal therapy elicited no evidence of toxicity (Supplementary Fig. 7b, c). Thus, in multiple tumor models immunoliposome therapy maintained the anti-tumor efficacy but eliminated multiple features of the systemic toxicity elicited by free IL-2-Fc + anti-CD137 treatment.

Melanoma is an aggressive disease that is highly lethal upon metastasis. We thus also tested the anti-tumor efficacy of immunoliposome therapy in a lung metastasis model. Six days post i.v. injection of luciferase-expressing B16F10 cells, when pulmonary tumor nodules could be clearly detected by bioluminescence imaging, treatment was initiated. In untreated and Lipo-IgG control groups, growth of lung tumors escalated quickly 2 weeks after tumor cell inoculation (Fig. 5a). In striking contrast, Lipo-αCD137/IL-2-Fc therapy greatly suppressed tumor growth, with negligible lung signal detected through ~25 days, as monitored by IVIS bioluminescence imaging (Fig. 5a, b). Following the third treatment on day 10, mice administered Lipo-αCD137/IL2-Fc already showed a significant reduction of tumor burden in the lung comparing to control Lipo-IgG and untreated groups (Fig. 5b, c). By contrast, soluble anti-CD137/IL-2-Fc failed to elicit any anti-tumor effect, and all animals treated with the soluble combination died on day 10 due to accumulated in vivo toxicity (Fig. 5b–d). Lipo-αCD137/IL-2-Fc significantly suppressed tumor progression in the lung and enhanced survival comparing to untreated and Lipo-IgG groups, with 20% of animals achieving complete tumor eradication (Fig. 5e).

Inspired by the striking therapeutic efficacy in the early treatment model, we further assessed whether Lipo-αCD137/IL-2-Fc was effective in treating metastases at a later stage. Treatment with three injections of the αCD137/IL-2-Fc combination was initiated on day 14 post tumor injection, when much larger pulmonary metastatic foci were established compared to day 6. Free αCD137/IL-2-Fc failed to control tumor growth and

led to lethal immunotoxicity as before, but mice treated with Lipo-αCD137/IL-2-Fc showed no signs of toxicity and had a significantly lower tumor burden in the lungs than all other groups (Fig. 5f). Therefore, in the lung metastasis model, Lipo-αCD137/IL-2-Fc therapy could eradicate tumors in a fraction of animals when therapy was administered at an early stage and also significantly inhibited tumor growth in the lungs even the therapy was initiated at later stages of tumor progression. Similar to the results in treatment of flank tumors, lethal toxicity accompanying the soluble form of this combination therapy was eliminated by immunoliposome delivery.

**IL-2-Fc plays a dominant role in therapy toxicity**. To better understand the side effects induced by αCD137/IL-2-Fc treatment, we exploited the fact that liposome delivery blocks the side effects of this therapy to determine the relative contributions of the two agents to the toxicity of the soluble therapy. To this end, combination treatments were administered where one of the agents was delivered in soluble form and one was liposome-bound and serum cytokines and chemokines were evaluated after a single injection. As before, soluble αCD137 + IL-2-Fc triggered significant elevations of IFN-γ, IL-6, MCP-1, and TNF-α (Fig. 6a–d). Liposomal delivery of αCD137 with soluble IL-2-Fc also induced inflammatory cytokines and chemokines, at levels one-third to one-half that seen with soluble αCD137/IL-2-Fc treatment. By contrast, Lipo-IL-2-Fc combined with soluble αCD137 induced minimal or undetectable levels of inflammatory cytokines comparable to control untreated animals, suggesting that the bulk of the toxicity derived from the soluble combination derives from the long half-life IL-2 molecule (Fig. 6a–d).

**Liposomes induce intratumoural immune responses**. Given the effective accumulation of the immunoliposomes in tumors and their substantial anti-tumor efficacy, we hypothesized that the immunoliposomes effectively stimulate an intratumoural immune response despite lowering systemic exposure to the two immunomodulators. To test this idea, we evaluated immunologic changes in the tumor microenvironment following systemic anti-CD137 and IL-2-Fc combination therapy. One day after three injections of αCD137/IL-2-Fc therapy in tumor-bearing mice, tumors and blood were collected and subjected to flow cytometry analysis for lymphocytes counts, cytokine production, and biomarker analysis (Fig. 7a). Both αCD137/IL-2-Fc and Lipo-αCD137/Lipo-IL-2-Fc treatment induced substantial CD8+ T-cell and NK cell tumor infiltrates (to comparable levels) and increased intratumoral CD8+/Treg ratios approximately fivefold (Fig. 7b–d). We further evaluated whether T-cells in tumors, blood and TDLNs were primed following treatment, by analyzing IFN-γ production in response to ex vivo polyclonal restimulation. We found increased numbers of IFN-γ-producing CD8+ and CD4+ T-cells in both cohorts receiving αCD137/IL-2-Fc compared to control Lipo-IgG and untreated groups (Fig. 7e–g), with soluble and liposome therapy again eliciting similar levels of T-cell activation. Additionally, we evaluated T-cell degranulation and T-cell cytotoxicity upon ex vivo polyclonal stimulation. Lipo-αCD137/IL-2-Fc and αCD137/IL-2-Fc induced a two to threefold elevation of CD107 expression in CD8+ T-cells from tumors (Fig. 7h) as well as elevated granzyme B expression in CD8+ T-cells in TILs (Fig. 7i). Altogether these results suggest that both soluble and liposome-delivered αCD137/IL-2-Fc therapy strongly induces T-cell priming in tumors and TDLNs, and promotes a pro-immune shift in the composition of intratumoral leukocytes. To obtain some insight into the duration of stimulation following liposome treatment, 24 h after three i.v. injections of soluble or liposomal immunotherapy, we assayed for levels of

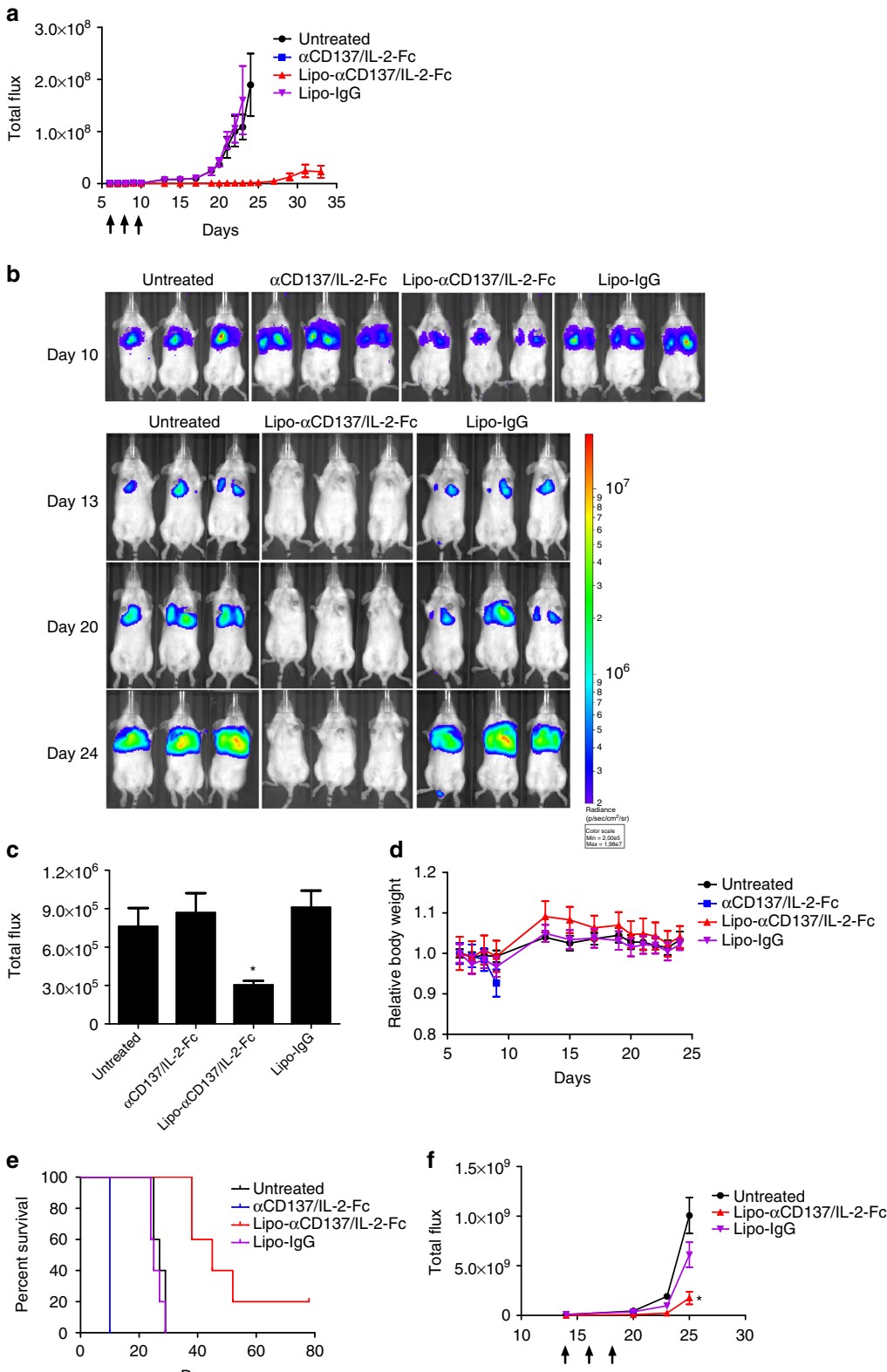

**Fig. 5** Immunoliposome therapy suppressed tumor progression in a lung metastatic model with low toxicity. **a–e** Groups of C57Bl/6 mice ($n = 5$ per group) were injected on day 0 with $5 \times 10^5$ luciferase-expressing B16F10 tumor cells i.v. Animals then received i.v. injections of αCD137 + IL-2-Fc, Lipo-αCD137 + Lipo-IL-2-Fc, or Lipo-IgG on days 6, 8, and 10 (100 µg per dose αCD137 and 60 µg per dose IL-2-Fc). **a** Mean total flux of lung luciferase signal from mice vs. time. **b** Bioluminescence images during the therapy. **c** Comparison of mean luciferase signals on day 10. *$p < 0.05$ Lipo-CD137/IL-2-Fc vs. untreated, αCD137/IL-2-Fc and Lipo-IgG. **d** Mean relative body weights of animals during the therapy (normalized to day 6). **e** Overall survival. **f** Groups of C57Bl/6 mice ($n = 5$ per group) were injected on day 0 with $5 \times 10^5$ luciferase-expressing B16F10 tumor cells then received i.v. injections of αCD137/IL-2-Fc, Lipo-αCD137/IL-2-Fc or Lipo- IgG on days 14, 16, and 18 (100 µg/dose αCD137 and 60 µg/dose IL-2-Fc). Shown are mean lung luciferase signals vs. time. *$p < 0.05$, Lipo-CD137/IL-2-Fc vs. untreated and Lipo-IgG. All measurements shown are mean ± s.e.m.

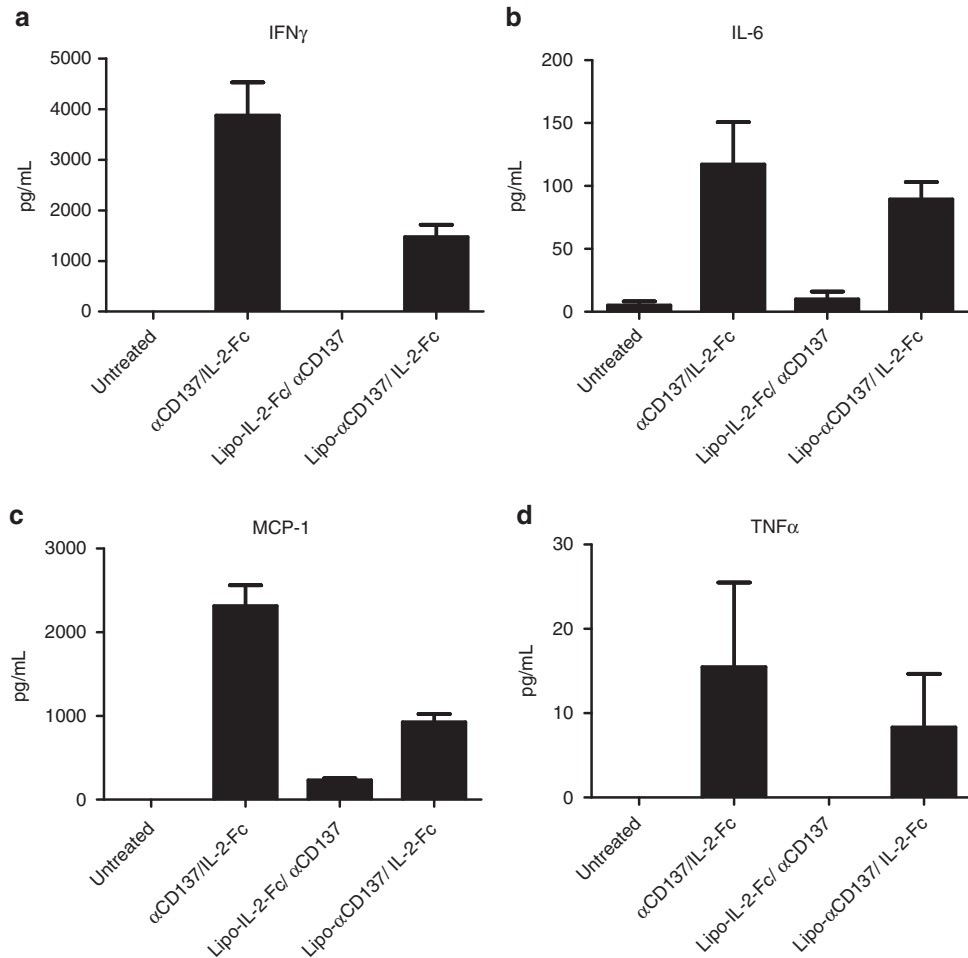

**Fig. 6** Systemic inflammatory toxicity mainly derived from IL-2-Fc rather than anti-CD137. Groups of C57Bl/6 mice ($n = 4$/group) were inoculated with $5 \times 10^5$ B16F10 tumor cells s.c. on day 0, then received i.v. injections of αCD137 and IL-2-Fc (αCD137/IL-2-Fc), Lipo-IL-2-Fc mixed with αCD137 (Lipo-IL-2-Fc/αCD137), or Lipo-αCD137 mixed with IL-2-Fc (Lipo- αCD137/IL-2-Fc) on day 8. One day later, sera were collected from peripheral blood and levels of IFNγ (**a**), IL-6 (**b**), MCP-1 (**c**) and TNFα (**d**) were assayed by luminex assays. All measurements shown are mean ± s.e.m.

phosphorylated STAT5 in intratumoural T-cells and NK cells (a downstream product of IL-2 receptor signaling). Both soluble and liposomal αCD137/IL-2-Fc induced intracellular *p*STAT5 activation in CD8 T-cells and NK cells in both primary (Fig. 7j) and lung metastatic (Fig. 7k) melanoma models, with the soluble agents activating pSTAT5 in NK cells slightly more effectively than liposomes. These results suggest that both soluble and liposomal immune-agonist signaling persists for at least 24 h post administration.

## Discussion

Cytokines and immune-agonist antibodies, while of great interest for cancer immunotherapy, often result in the development of severe dose-limiting side effects, limiting their therapeutic potential. In the case of cytokines, these side effects are in part linked to the high doses that are often administered to compensate for the very short half-lives these small proteins tend to exhibit in the circulation. For example, high-dose (HD) IL-2 cancer immunotherapy stimulates the proliferation of cytotoxic CD8[+] T-cells and NK cells, promoting tumor regression[24,25], and it has been approved for the treatment of metastatic melanoma and renal cancer for its anti-tumor activity[26,27]. However, HD IL-2 therapy is accompanied by a systemic cytokine storm and excessive vascular permeability known as VLS, and thus intensive patient management is required during IL-2 therapy[26]. These

undesirable adverse effects are life-threatening and result in the compromised response to immunotherapies.

These challenges have motivated a number of strategies to improve the safety profiles while maintaining the therapeutic efficacy of immune agonists, particularly cytokines. Cytokines have been molecularly engineered to alter their specificity for receptor subunits in order to restrict their stimulation to only certain cellular targets[28], or complexed with monoclonal antibodies to achieve a similar goal[15]. Approaches such as PEGylation have been used to extend the half-life of cytokines in order to allow lower cytokine dosing, but the therapeutic window of these approaches remains narrow[29,30]. Targeting of the action of cytokines to the tumor microenvironment should provide another approach to minimize systemic toxicity from immune agonists, and this concept has been pursued clinically in the form of immunocytokines comprised of antibodies fused with cytokine components[31]. However, systemic administration of these engineered agonists often fails to achieve targeted tumor delivery, due to the rapid binding of the cytokine component to circulating immune cells[32]. Thus, novel strategies to target immune-agonist function to tumors remain of interest.

IL-2 and agonistic anti-CD137 act in complementary pathways to promote multiple aspects of innate and adaptive immunity against tumors, but exhibit significant toxicity as monotherapies in humans and are highly toxic as a combination treatment in mice. Use of IL-2-Fc fusions to increase the half-life of IL-2

increase anti-tumor efficacy but still induce substantial toxicity at functional doses. We previously showed that conjugation of IL-2 and anti-CD137 to liposomes for intratumoural injection controlled tumor progression without toxicity[11], demonstrating that IL-2/anti-CD137 signaling confined to tumors and TDLNs is sufficient for anti-tumor efficacy, suggesting that the toxicity arising from systemic exposure to these agents is separable from

anti-tumor activity. However, strategies enabling systemic treatment of disseminated metastatic disease are needed.

Here we show that an approach well-studied in the nanomedicine field, using nanoparticle as a carrier to promote tumor accumulation of these drugs, is particularly effective as an approach for immunotherapy. Macromolecules and nanoparticles administered systemically accumulate in tumors via the EPR

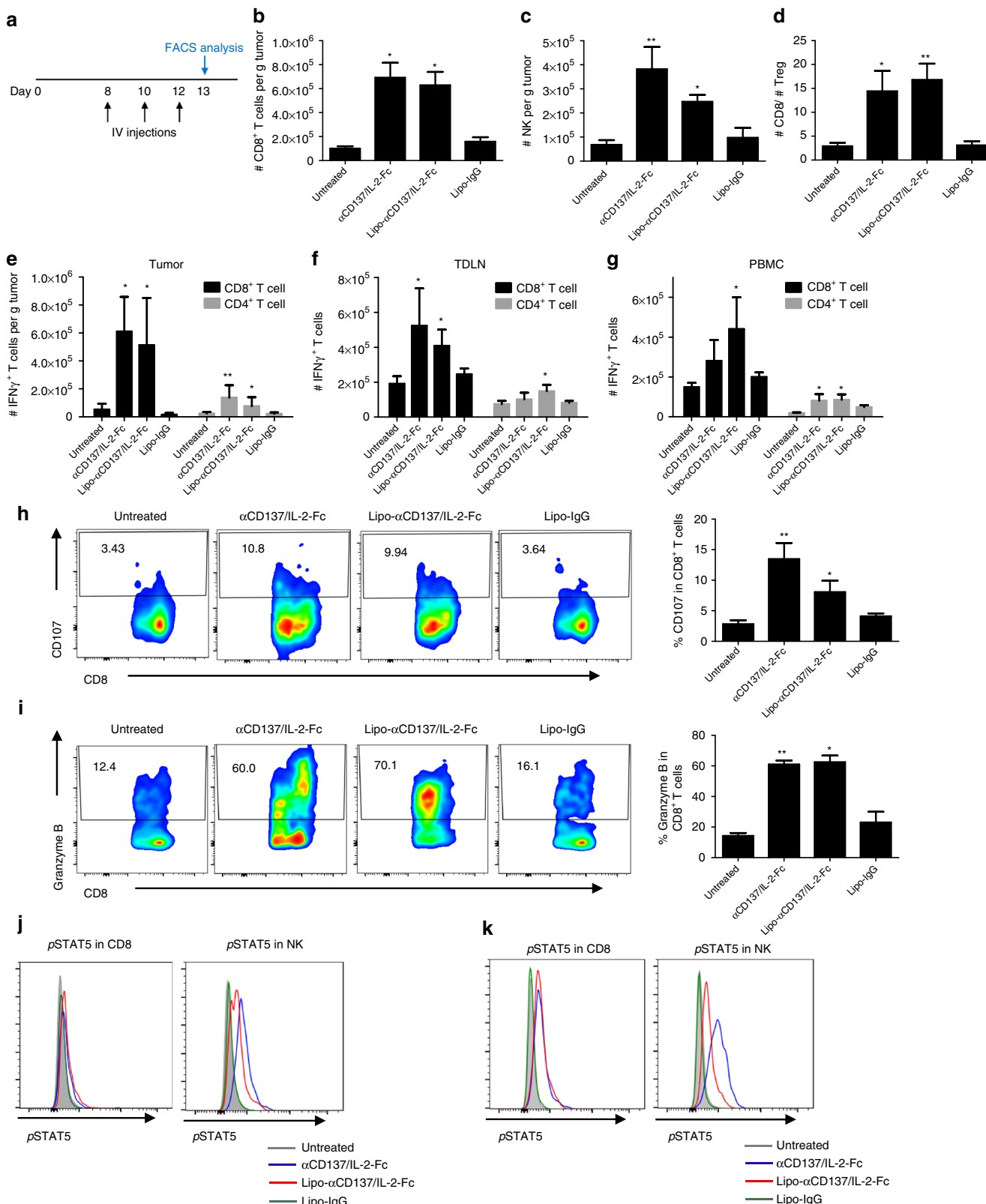

effect, whereby leaky tumor vasculature and poor lymphatic clearance in tumors lead to accumulation of these agents in tumors over time[33]. While wild-type IL-2 has a very short half-life precluding significant tumor accumulation following a single systemic administration[34], extended-PK IL-2-Fc circulates long enough to accumulate in tumors, together with the long half-life anti-CD137 antibody. However, antibody-sized macromolecules accumulate slowly in tumors, with peak accumulation not occurring until ~48 h after administration[35]. Thus, for these free protein drugs, it is impossible to achieve efficient tumor accumulation without extended systemic exposure. Contrasting to the slow buildup of therapeutic proteins in tumors, nanoparticles in the 50–200 nm size range show much more rapid tumor accumulation; in studies where early time points have been measured, maximal tumor accumulation has been observed as early as 4–6 h after administration[36,37]. This timescale also suggests that nanoparticles with much shorter half-lives than antibodies can effectively accumulate in tumors with a much lower total systemic exposure. We covalently linked IL-2 and anti-CD137 to liposomes, rather than encapsulating free forms of these drugs, to avoid release of the free molecules, which as shown by our prior studies of injecting soluble IL-2/anti-CD137 leads to systemic exposure and cytokine storms even following intratumoral injection[11]. By covalently linking the payloads to the liposome surfaces, their biodistribution is dominated by the liposome trafficking and release of free cytokine/antibody is minimal.

Using PEGylated liposomes as a scaffold for delivery of covalently anchored anti-CD137 and IL-2-Fc, we showed that these nanocarriers modulated the pharmacokinetics and biodistribution of the toxicity-inducing cytokines, providing rapid accumulation in tumors with reduced systemic exposure. Modest alterations in the systemic circulation behavior correlated with an elimination of systemic cytokine storms, VLS, and lethal toxicity. However, the nanoparticle forms retained the full anti-tumor activity of the parent drug combination, and achieved similar priming of immune cell infiltration and T-cell activation within tumors as the free cytokine/antibody combination.

A strength of the approach described here is that it should be generalizable to diverse immunomodulatory agents including small molecule and biologic drugs, and immunoliposomes have already reached clinical testing in the setting of targeting chemotherapy drugs to tumors. We demonstrate here that these platforms, even without encapsulated drug cargos, can be very effective when the particles carry immune agonists rather than tumor-targeting moieties, relying on the EPR effect for a small but effective dose of these drug to reach tumors following systemic administration. While it continues to be debated how homogeneous and effective the EPR effect is in human patients[38,39], there are several points that make us encouraged that the data shown here in mice could predict efficacy in

humans. First is the key distinction between the drug agents/cellular targets of traditional nanomedicine—where typically chemotherapy or targeted drugs must reach tumor cells, and must deliver a functional dose of drug to every single tumor cell—vs. immunotherapy, where small amounts of potent cytokines/antibodies reaching a proportion of immune cells in the tumor microenvironment may have efficacy, or which may bind immune cells in circulation that carry these agents into the tumor. Notably, our data in a lung metastasis melanoma model demonstrated that immunoliposomes could elicit superior therapeutic efficacy and survival benefits for treating metastatic lesions compared to their soluble counterparts, in both small metastatic lesions and larger established metastatic nodules. Second, imaging-based clinical methods (e.g., magnetic resonance imaging imaging of iron oxide nanoparticles to identify patients exhibiting nanoparticle accumulation in tumors prior to nanoparticle therapy), are beginning to be employed clinically[40,41], which pave the way for rational selection of patients for EPR-dependent therapies. Lastly, a variety of clinically-relevant strategies to increase the EPR effect in tumors are beginning to be defined, such as radiation therapy[42,43], photodynamic therapy[44], and use of vasodilators[45], to name just a few. We hypothesize that this approach is better suited to immunostimulatory agents that act at low doses rather than drugs that block inhibitory receptors (e.g., anti-PD-1, anti-CTLA-4), which may require high doses to fully block target receptors on tumor-infiltrating immune cells. Further work will be required to evaluate these issues.

## Methods

**Materials.** Anti-CD137 (clone LOB12.3), anti-CD8a (clone 2.43), anti-NK1.1 (clone PK136), and rat IgG isotype control antibodies (clone HRPN) were from BioXCell. Dioleoylphosphocholine (DOPC), polyethylene glycol 2000-distearoylphosphoethanolamine (DSPE-PEG), and maleimide-PEG2000-DSPE were obtained from Avanti Polar Lipids. Fluorescent antibodies against mouse CD45 (1:100 dilution, cat# 17–0451–82), CD3 (1:100 dilution, cat# 48–0032–82), CD4 (1:100 dilution, cat# 45–0042–82), NK1.1 (1:100 dilution, cat# 25–5941–81), IFNγ (1:50 dilution, cat# 12–7311–41), CD16/32 (1:100 dilution, cat# 14–0161–82), CD25 (1:100 dilution, cat# 45–0251–82) and Foxp3 (1:50 dilution, cat# 17–5773–82) were from eBioscience. Anti-mouse CD8a (1:100 dilution, cat# 100747), CD107a (1:100 dilution, cat# 121606), CD107b (1:100 dilution, cat# 108504), granzyme B (1:25 dilution, cat# 515406) and FOXP3 Fix/Perm buffer set were from BioLegend. Luminex multiplex magnetic bead kits and centrifugal filter units were obtained from EMD Millipore. Alexa fluorophore dyes, Pierce F(ab')₂ preparation Kit and Dynabeads mouse T-Activator CD3/CD28 for T-cell expansion and activation kits and LIVE/DEAD Fixable Aqua Dead Cell Stain kits were purchased from Thermo Fisher Scientific. IRDye 800CW Infrared dye was from LI-COR. Alexa Fluor 488-labeled ovalbumin (OVA) was from Invitrogen. Concanavalin A Type VI (Con A) and cholesterol were from Sigma-Aldrich. Recombinant IL-2 and IL-7 were purchased from PeproTech. EasySep Mouse CD8+ T-cell enrichment Kit was from Stemcell. D-luciferin was from Caliper Life Sciences. Collagenase IV and DNase were from Roche. FTY720 was purchased from Cayman Chemical.

**Immunoliposome preparation and characterization.** A lipid mixture composed of DOPC, cholesterol, DSPE-PEG, and DSPE-PEG-maleimide in a molar ratio of

**Fig. 7** Anti-CD137/IL-2-Fc therapy induces intratumoural immune activation. Groups of C57Bl/6 mice (n = 5 per group) were injected on day 0 with 5 × 10⁵ B16F10 tumor cells s.c. Animals then received i.v. injections of αCD137 + IL-2-Fc, Lipo-αCD137 + Lipo-IL-2-Fc, or Lipo-IgG on days 8, 10, and 12 post tumor cell inoculation (100 µg per dose αCD137 and 60 µg per dose IL-2-Fc). On day 13, tumors and blood were processed to single cell suspensions for flow cytometry analysis. **a** Timeline of i.v. injections and flow cytometry analysis. **b, c** The amounts of tumor-infiltrating CD8+ T-cells (**b**) and NK cells (**c**). **d** The ratio of CD8+ T-cells and regulatory T-cells (Tregs) in tumors. *p < 0.05, **p < 0.005. **e–g** Lymphocytes from the tumors (**e**), TDLN (**f**), and blood (**g**) were polyclonal stimulated in vitro. IFN-γ production by T-cells was analyzed by intracellular cytokine staining. **p < 0.005, *p < 0.05 vs. Untreated. Lymphocytes from tumors were polyclonal stimulated in vitro with the presence of CD107a and CD107b antibodies. **h** Representative FACS plots and statistical analysis of CD107 expressions on CD8+ T-cells in tumors. The production of granzyme B by CD8+ T lymphocytes was analyzed by intracellular staining, as the indication of cytotoxic killing effect of activated T-cells by degranulation. *p < 0.05, **p < 0.01. **i** Representative flow cytometry plots and statistical analysis of granzyme B expressions in CD8+ T-cells in tumors. *p < 0.005, **p < 0.001. **j, k** Intracellular pSTAT5 activation in CD8 T-cells and NK cells in tumor sites of primary melanoma (**j**) and lung metastatic melanoma (**k**) after the aforementioned treatments. Gray: untreated, blue: αCD137/IL2-Fc, red: Lipo-αCD137/IL2-Fc, turquoise: Lipo-IgG. All measurements shown are mean ± s.e.m.

47/49/2/2 mol% was vacuum-dried in a glass vial and rehydrated in pre-warmed PBS with vigorous vortexing for 1 min, followed by bath sonication at 50 °C for 5 min The resulting liposomes were pelleted by ultracentrifugation (Beckman Coulter) at 40,000 r.p.m. for 4 h at 4 °C and the supernatant was removed. The liposome pellet was resuspended in PBS prior to antibody/IL-2 conjugation. IL-2-Fc is a bivalent protein of murine wild-type IL-2 linked to a mouse IgG2a backbone and was produced as previously described[11]. A D265A mutation was introduced in the IgG2a Fc region to minimize the interaction with Fc receptors on phagocytes. To avoid opsonization by phagocytes, the Fc portion of anti-CD137 was digested by pepsin and cleaved by protein A using a Pierce F(Ab′)2 preparation Kit before coupling to liposome surfaces. For experiments involving labeled proteins, 10 µg Alexa Fluor dye 568 NHS Ester or IRDye 800CW NHS ester was added to 100 µg anti-CD137 or IL-2-Fc (total concentration 2 mg/mL in 10 mM K2HPO4 pH 9) and mixed for 20 min, followed by washing away unreacted dye molecules using a centrifugal filter unit with a 30 K MWCO membrane. For coupling IL-2-Fc and anti-CD137 FAb2 proteins to maleimide-functionalize liposomes, antibody or IL-2-Fc were reduced by 1.8 mM dithiothreitol (DTT) in the presence of 10 mM EDTA for 20 min at 25 °C to expose free thiols in the IgG framework hinge region. DTT was then removed by a centrifugal filter unit with 30 K MWCO membrane. Fluorescently labeled antibodies were reduced in the same way. The reduced anti-CD137, IL-2-Fc, or isotype control antibodies (final concentration ~10 µg/µL) were immediately added to maleimide-bearing liposomes (5 mg/mL lipids) in PBS and allowed to react for 18 h at room temperature with continuous mixing. β-mercaptoethanol (0.5 mM) was then added to block unreacted maleimide. The resulting immunoliposomes were pelleted by ultracentrifugation and the supernatant containing unreacted antibodies was removed. The pelleted antibody-conjugated immunoliposomes were resuspended in PBS and stored at 4 °C until use.

Hydrodynamic sizes and zeta potentials of immunoliposomes were measured by DLS (Malvern Zetasizer Nano). Cryo-TEM was performed using a JEOL 2100F transmission electron microscope. Fluorescently labeled antibodies were used in the PK studies (Fig. 2d, e), biodistribution studies (Fig. 2f–i), confocal tissue imaging (Fig. 3), and some flow cytometry analysis (Fig. 2j), and allowed direct determination of the amount of conjugated anti-CD137/IL-2-Fc. In experiments where the cytokine/antibody payloads were not fluorescently labeled, the antibody loading was determined by an infrared spectroscopy protein assay, using a Direct Detect infrared spectrometer (EMD Millipore).

For the immunoliposome in vitro stability assay, Alexa Fluor 488-labeled OVA was encapsulated in liposome as a tracer of the liposome. Briefly, 1.4 mg dried lipids were hydrated in pre-warm PBS containing 46 µg Alexa Fluor 488-labeled OVA with vigorous vortexing for 1 min and bath sonication at 50 °C for 5 min, followed by manual extrusion through 0.1 µm pore size polycarbonate membrane. The resulting liposomes were pelleted by ultracentrifugation at 50,000 rpm for 4 h at 4 °C and the supernatant was removed. Anti-CD137 and IL-2-Fc were conjugated with IRDye 800CW fluorescent dye prior to coupling to OVA-loaded liposomes. The ratios of anti-CD137 or IL-2-Fc to liposome (OVA fluorescence) over time were measured in vitro at 37 °C with continuous rotating, using PBS containing 10% FBS as the release medium. At predetermined time points, the release medium containing fluorescence labeled immunoliposomes was subjected to ultracentrifugation at 50,000 r.p.m. at 4 °C for 4 h. The resulting immunoliposome pellet was disrupted by Triton X-100, and the fluorescence intensities of Alexa Fluor 488 and IRDye 800CW were measured by a fluorescence plate reader and LI-COR infrared spectroscopy instrument, respectively. Normalized to the initial time point (set ratio as 1), the relative ratios of anti-CD137 or IL-2-Fc to liposome fluorescence were calculated.

**Cell lines**. B16F10 and A20 cells were from American Type Culture Collection (ATCC), and B16F10-FLuc cell was made by transfection with a CMV promoter-driven Firefly luciferase reporter gene expression plasmid vector. The Trp2-KO B16F10 cell line (B16F10-Trp2KO) was generated using CRISPR-Cas9 technology[3]. B16F10, B16F10-FLuc, and B16F10-Trp2KO cells were cultured in DMEM (high glucose) medium supplemented with 10% fetal bovine serum (FBS) and antibiotics containing penicillin and streptomycin. A20 cells were cultured in RPMI-1640 medium supplemented with 10% FBS and antibiotics. Growth medium was changed every 2–3 days.

**Animals and tumor therapy**. The experimental handling of mice was conducted under federal, state, and local guidelines and following an approved protocol from the Division of Comparative Medicine (DCM) of the Massachusetts Institute of Technology. Except where otherwise noted, C57BL/6 mice (6–8-week-old female, the Jackson laboratory) were subcutaneously inoculated with $5 \times 10^5$ B16F10 cells, and treated on indicated days by i.v. injection of anti-CD137 and IL-2-Fc or other combinations as noted. For the A20 B cell lymphoma model, female Balb/c mice were subcutaneously inoculated with $2 \times 10^6$ A20 cells.

Lung metastases were established in C57BL/6 mice by i.v. injection of $5 \times 10^5$ B16F10-FLuc cells, which led to the formation of pulmonary metastatic tumor foci in the lung. Tumors were traced following inoculation by bioluminescence imaging. The total flux obtained from bioluminescence imaging over a fixed thoracic area was used as a quantitative measurement of lung tumor burden.

Animals were treated by i.v. injection of αCD137/IL-2-Fc combinations as noted in the text.

For subcutaneous tumors, tumor size was monitored every other day with calipers, and areas were calculated as the product of orthogonal diameters of the tumor. In the lung metastasis model, tumor burden was measured by bioluminescence imaging with a Xenogen IVIS Spectrum Imaging System. Each mouse received 150 mg luciferin/kg body weight subcutaneously 10 min before imaging. Data was analyzed by Living Imaging Software. Body weight of the treated mice were monitored as an indication of in vivo toxicity. Relative body weight was normalized to the initial day of treatment. The physical condition and survival of mice from each treatment group were monitored daily during the study.

**In vitro T-cell binding and stimulation assays**. Splenocytes from C57Bl/6 mice were activated by conconavalin A (2 µg/mL) and IL-7 (1 ng/mL) for 2 days, and the activated CD8+ T-cells were isolated using an EasySep Mouse CD8+ T-cell enrichment Kit. To assess cellular binding, Alexa Fluor 555-conjugated Lipo-IL-2-Fc, Lipo-αCD137, or Lipo-IgG were incubated with activated CD8+ T-cells for 3 h at 37 °C, followed by analysis using flow cytometry and confocal microscopy. For bioactivity analysis, T-cells from C57BL/6 mice were activated by CD3/CD28 beads (1:1 bead:cells ratio) for 2 days before the addition of IL-2-Fc or Lipo-IL-2-Fc (final IL-2 concentration 10 ng/mL). CD8+ T-cell and CD4+ T-cell counts were analyzed by flow cytometry on day 3 and day 4 post activation. Activated T-cells were incubated with anti-CD137 or Lipo-αCD137 (final anti-CD137 concentration 10 µg/mL) in the presence of 10 ng/mL recombinant murine IL-2 for 6 h, and the secreted IFN-γ in the cell culture supernatant was analyzed by ELISA.

**Pharmacokinetics study**. Alexa Fluor647-conjugated Anti-CD137, IL-2-Fc, or isotype control IgG were coupled to liposomes as described above. B16F10 tumor-bearing mice were then given i.v. injections of the fluorescently labeled anti-CD137, Lipo-αCD137, IL-2-Fc, or Lipo-IL-2-Fc. At predetermined time points, 20 µL blood from each mouse was collected via retro-orbital bleeding with a heparin-coated capillary. The fluorescence intensities in sera were measured by a Typhoon FLA 7000 fluorescence spectrometer and the concentration of antibody was calculated from a standard curve prepared for each reagent. The antibody concentrations in sera were fit to a bi-phasic exponential decay: MFI $(t) = Ae^{-\alpha t} + Be^{-\beta t}$. Pharmacokinetic parameters were calculated. $t_{1/2,\alpha}$, fast clearance half-life; $t_{1/2,\beta}$, slow clearance half-life; AUC, area under the curve.

**Biodistribution analysis**. IRDye 800CW infrared dye-conjugated anti-CD137, IL-2-Fc, or isotype control IgG were coupled to liposomes as described above. B16F10 tumor-bearing mice then received i.v. injections of the infrared dye labeled anti-CD137, Lipo-αCD137, IL-2-Fc, or Lipo-IL-2-Fc. Four hours and 24 hours later, blood was drawn, and organs were collected and weighed. The fluorescence intensities in blood and all organs were measured by a LI-COR Odyssey imaging system, and a percentage (%) of injected dose accumulated in organs was calculated from predetermined standard curves, normalized by organ weights. For preparing the standard curves, organs were dissected from mice and known doses (0, 0.0016, 0.008, 0.04, 0.2 µg) of IRDye CW800 infrared dye were injected into individual organs before the fluorescence intensities were measured using a LI-COR Odyssey imaging system. The standard curve equations and R-squared ($R^2$) values were determined by Excel and GraphPad Prism software.

**Luminex assay**. Blood was collected from treated mice via retro-orbital bleeding on indicated days and levels of inflammatory cytokines and chemokines (IFNγ, IL-6, MCP-1, TNFα) in serum samples were determined by a Luminex assay using a Luminex Milliplex Magnetic Bead Kit (EMD Millipore).

**Lymphocyte depletions**. B16F10 tumor-bearing mice were given intraperitoneal (i.p.) injections of FTY720 at a dose of 3 mg/kg. One day later, mice were intravenously treated with anti-CD137 and IL-2-Fc, with 100 µg per dose of anti-CD137 and 60 µg per dose of IL-2-Fc. Animals subsequently received daily i.p. injections of 1 mg/kg FTY720 thereafter. One day and 3 days after anti-CD137/IL-2-Fc injection, peripheral blood was drawn and sera were collected. Cytokine and chemokine levels in sera were measured by luminex cytokine assays as described above. For NK cell and CD8+ T-cell depletion studies, B16F10 tumor-bearing mice were given i.p. injections of 400 µg NK1.1 antibody, 400 µg CD8a antibody, or isotype control antibody starting on day 6 post tumor cell inoculation, and the i.p. injections of these depleting antibodies were given every 4 days till the end of the study. B16F10 tumor-bearing mice received i.v. injections of anti-CD137 and IL-2-Fc on days 8, 10, and 13 post tumor cell inoculation, with 100 µg per dose anti-CD137 and 60 µg per dose IL-2-Fc.

**Flow cytometry analysis**. Mice bearing subcutaneous B16F10 melanomas were given i.v. injections of anti-CD137 and IL-2-Fc, Lipo-αCD137/IL-2-Fc or Lipo-IgG on days 8, 10, and 12 after tumor cell inoculation, with 100 µg per dose of αCD137 and 60 µg per dose of IL-2-Fc. Mice bearing B16F10-Fluc lung metastases were given the same treatment and dosing on days 6, 8, and 10. One day after the third injection, PBMC, TDLNs, tumors, or lungs were collected and processed to single

cell suspension for antibody staining and flow cytometry analysis on a BD LSRFortessa Cell Analyzer. Regulatory T-cell (Treg) levels were quantified by intracellular Foxp3 staining. Lungs were digested with collagenase IV and DNase for 30 min at 37 °C to extract lymphocytes. Cells from PBMC, TDLNs, tumors, or lungs were stimulated in vitro for 6 h with 12-myristate 13-acetate (PMA) (81 nM) and ionomycin (1.34 μM) in the presence of brefeldin A (3 μg/mL) for 4 h to block cytokine secretion. IFN-γ production by stimulated T-cells was measured by intracellular cytokine staining, as an indicator of T-cell effector function. In parallel, lymphocytes from tumors were polyclonally stimulated in vitro by PMA and ionomycin in the presence of CD107a and CD107b antibodies. One hour later, menosin and brefeldin A were added to block cytokine secretion, and continuously incubated for 4 h before antibody staining. The production of granzyme B by CD8[+] T lymphocytes were analyzed by intracellular staining, as an indication of cytotoxic killing effect of activated T-cells.

**Immunohistochemistry analysis**. Mice bearing subcutaneous B16F10 tumors were administered i.v. injections of anti-CD137 and IL-2-Fc, Lipo-αCD137/IL-2-Fc, or IL-2-Fc. One hour later, subcutaneous tumors were dissected and immediately snap-frozen in dry ice with OCT cryo-embedding media for cryosectioning. Ten micron tissue sections were cut for immunostaining procedures as follows. The slides were fixed twice in 100% ethanol for 30 min each time before permeabilized with 0.2% Triton X-100 for 10 min The slides were blocked with 5% goat serum for 1 h, followed by incubating with CD31 antibody (1:100) and DAPI (1:2000) diluted in 1% goat serum for 2 h. Then slides were washed four times with PBS before covering the slides with mounting medium. The images were taken under an Olympus FV1200 Laser Scanning Confocal Microscope, with a ×30 magnification. The images and quantitative analysis were processed using ImageJ software.

**Vascular leak syndrome measurement**. Mice bearing subcutaneous tumors were given two i.v. injections of αCD137/IL-2-Fc and Lipo-αCD137/IL-2-Fc, at 100 μg per dose anti-CD137 and 60 μg per dose IL-2-Fc. One day later, lungs were collected and weighed before lyophilization. The lyophilized lungs were then weighed. The water content (g) in the lung was calculated as: lung weight before lyophilization (g)—lung weight after lyophilization (g).

**Hematological test**. Serum was isolated from mice 1 day after two treatments. The liver enzymes AST and ALT, and cholesterol and creatinine kinase were analyzed in the MIT Diagnostic lab, according to the manufacturer's protocol and hospital standards.

**Statistical analysis**. One-way analysis of variance was used to identify the differences of measured continuous variables between groups. A two-tailed Student's $t$ test will be used for the comparison between two groups with equal variance. Survival analysis will be conducted using Kaplan–Meier curves and log-rank (Mantel-Cox) test. Statistical significance level will be defined as $P < 0.05$.

**Data availability**. All relevant data are available from the authors on request and are included with the manuscript and its supplementary information files.

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

## Acknowledgements

We thank the Koch Institute Swanson Biotechnology Center for technical support, specifically the flow cytometry core facility, the animal imaging and preclinical testing core facility, the histology core facility, and the nanotechnology materials core facility. This work was supported in part by the Koch Institute Cancer Center Support Grant P30-CA14051 from the National Cancer Institute. D.J.I. is an investigator of the Howard Hughes Medical Institute.

## Author contributions

Y.Z. performed all the experiments, analyzed the data, and wrote the manuscript. N.L. assisted with experiments. H.S. assisted with the production of IL-2-Fc. D.J.I. supervised the studies. Y.Z. and D.J.I. wrote the manuscript.

## Additional information

**Competing interests:** D.J.I. and Y.Z. are inventors on patents relating to the immunoliposome technology licensed to Torque Therapeutics.

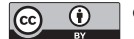

