## [Peer Review File · Nature Communications]

Reviewers' comments:

Reviewer #1 (Remarks to the Author):

This work, "Nanoparticle anchoring targets immune agonists to tumours enabling anti-cancer immunity without systemic toxicity", by Zhang and colleagues is an interesting and timely solution addressing the important problem of dose-limiting toxicities for immune therapy. The work is also an interesting platform for a diverse array of potential applications. I believe this work should be published in Nature Communications, although certain issues should be addressed prior:

1. It is not clear why the liposomes are formulated with surface-bound IL-2 and anti-CD-137 as opposed to packaged internally as a deliverable cargo? More rationale/discussion should be presented for the specific design of these particles. Alternatively, what might be the effect of having a single nanoparticle with both molecules tethered to the surface or integrated as cargo? What is the advantage of using a nanoparticle system? Would a polymer chain with both molecules (or separate polymer chains) achieve the same results, or might it penetrate the tumor better?

2. Greater consideration for the liposome stability, breakdown, and total surface presentation of immune molecules should be presented.

3. Biodistribution of the IV injected liposomes as presented in Figure 2 should also consider distribution to the tumor on a whole-animal level. The localized biodistribution as presented in Figure 3 is strong evidence, however it would be helpful to understand the % injected dose that localizes to the tumor (as in 2f and 2g). The authors make the statement that this can't be done due to melanin in the B16F10 model. However, it would be helpful analogous information to understand this with a subcutaneous tumor that does not produce melanin.

4. The experiments presented in Supplementary Fig. 7 are excellent controls for understanding the role of each molecule in toxicity and the role of the liposome. It may be useful to present this data in the main figures.

5. The authors rely on the EPR effect for localization of the nanoparticles to the tumors. As they discuss in the final paragraph of the discussion, the EPR effect may be heterogeneous or not present in various tumors and patients. If the rationale is that systemic localization of the immuno liposomes provides the ability to target metastases effectively and reduce off-target toxicities, it may be necessary to discuss the challenges of translating therapies to the clinic that rely on the EPR effect

more in-depth. It is also likely that the EPR effect plays even less of a role in small metastatic lesions and therefore the therapy may not be useful for micro-metastases. The role of the EPR effect in these small lesions should be discussed.

6. The fate of the nanoparticles in the tumor is unclear. It would be helpful to discuss how exactly they interact with the target cells. Are they internalized by cells? If so, how do the liposome-anchored molecules interact with the target cells in the tumor? How long is the particle active in the tumor?

7. It is unclear whether statistical analyses were performed for each individual figure panel (e.g. 1g, 2b, c, h, & 6d, h, i)

Reviewer #2 (Remarks to the Author):

This manuscript reports that using liposomes to deliver surface-anchored IL-2 and anti-CD137 proteins reduces the toxic side effects observed when these agents are delivered as soluble factors. The evaluation of the immunoliposomes includes multiple tumor models as well as evaluation of biodistribution and effect on TIL and resulting cytokine production. The potential significance of the work is high and the quality is generally quite good. However, there are a few issues that should be addressed before publication:

1. It is really unclear when immunoliposomes with fluorescently labeled antibodies were used, and in the experiments where they weren't used, how antibody loading was determined.
2. Biodistribution was determined by fluorescent labeling of immunoliposomes followed by whole organ fluorescence imaging using a Xenogen imager. The accuracy of this method should be demonstrated and verified, since it is non-traditional compared to direct radiolabeling and tissue quantification.
3. Importantly, no dosing studies are reported. It is very possible that the toxic side effects that are observed with the free proteins versus the immunoliposomes is simply because the free proteins were administered at too high a dose. In drug delivery, liposome delivery usually results in reduced side effects but also reduced efficacy; therefore investigators will often dose at a high dose so that efficacy is observed with their liposome formulation but toxicity is observed with the free drug. This is artificially biases the outcome to be favorable to the liposome formulation.

4. The extension to humans is not straightforward since the EPR effect can be strong in mouse models given the right tumor model, but is usually not observed as robustly in humans. This should be mentioned, since the advantages presented here depend on a strong EPR effect.

RESPONSE TO REVIEWS

Reviewer #1 (Remarks to the Author):

This work, “Nanoparticle anchoring targets immune agonists to tumours enabling anti-cancer immunity without systemic toxicity”, by Zhang and colleagues is an interesting and timely solution addressing the important problem of dose-limiting toxicities for immune therapy. The work is also an interesting platform for a diverse array of potential applications. I believe this work should be published in Nature Communications, although certain issues should be addressed prior:

We thank the referee for these positive comments.

1. It is not clear why the liposomes are formulated with surface-bound IL-2 and anti-CD-137 as opposed to packaged internally as a deliverable cargo? More rationale/discussion should be presented for the specific design of these particles. Alternatively, what might be the effect of having a single nanoparticle with both molecules tethered to the surface or integrated as cargo? What is the advantage of using a nanoparticle system? Would a polymer chain with both molecules (or separate polymer chains) achieve the same results, or might it penetrate the tumor better?

These are good questions and additional ideas raised by the referee. The IL-2 and anti-CD137 are presented on the particle surfaces to allow the drugs' biodistribution behavior to be governed by the particulate delivery form, and prevent their release in a “free” state that could lead to systemic exposure- if prepared in an encapsulated form for release from inside the liposomes, there is the potential for systemic exposure to the free drug even if release happens after the particles accumulate in the tumor microenvironment; we have previously shown (ref 11) that free IL-2 and anti-CD137 injected intratumorally will leak into the bloodstream and lead to systemic toxicity. Covalent anchoring to the liposomes prevents this possible systemic exposure. We would predict co-anchoring of both agents to the same liposome could be equally effective, and we only studied them in the separate state to allow easy titration of the dose of each molecule. The main role for the nanoparticle as noted above is to control the biodistribution of the two immunotherapy agents, and we would thus expect that linkage of these agents to high molecular weight polymer chains might be useful, though contingent on the rapid blood clearance behavior we observed here with liposomal anchors. We have added comments on these possibilities to the discussion on p. 14.

2. Greater consideration for the liposome stability, breakdown, and total surface presentation of immune molecules should be presented.

We performed additional experiments to measure the liposome stability and breakdown in the presence of serum (**new Supplementary Fig. 5**). To do this, liposomes were prepared encapsulating Alexa Fluor 488-labeled ovalbumin (OVA) as a tracer of intact vesicles, and anti-CD137 or IL-2-Fc labeled with an infrared dye were conjugated to the vesicle surfaces. These dual-labeled vesicles were then incubated in serum-containing PBS at 37°C *in vitro* for different time periods, followed by pelleting of the liposomes by ultracentrifugation and measurements of the liposome (OVA) to antibody/IL-2 ratio over time. As shown in Suppl. Fig. 5, the drug:liposome ratio remained approximately constant over 48 hr, suggesting the immune agonists were stably conjugated on the liposome surface over time. The total surface presentation of immune molecules was calculated according to the conjugation efficiency (%) of the drug

payloads, and showed that per 1 mg lipids, there were ~123 µg anti-CD137 or ~99 µg IL2-Fc conjugated on the liposome surfaces. These new data have been discussed in the revised text on pp. 6 and 7.

3. Biodistribution of the IV injected liposomes as presented in Figure 2 should also consider distribution to the tumor on a whole-animal level. The localized biodistribution as presented in Figure 3 is strong evidence, however it would be helpful to understand the % injected dose that localizes to the tumor (as in 2f and 2g). The authors make the statement that this can't be done due to melanin in the B16F10 model. However, it would be helpful analogous information to understand this with a subcutaneous tumor that does not produce melanin.

Since the melanin in B16F10 melanoma tends to block the fluorescence signal from the fluorophore conjugated on the liposomes, an unpigmented B16F10 Trp2 knock out cell line (B16F10-Trp2KO) that does not produce melanin (Moynihan et al. *Nat. Med.* 2016) was used as a substitute for B16F10 in the tumor accumulation experiment for the quantitation of the % injected dose that localized to the tumor. This additional data is shown in new Figs. 2h and 2i. In B16F10-Trp2KO tumours, liposomal anti-CD137 showed a ~5-fold increase in tumor accumulation compared to free anti-CD137 at 4 h and 24 h time points, while liposomal IL-2-Fc showed a ~50% increase of tumor accumulation comparing to that of free IL-2-Fc at 4 h, and the tumor accumulations of liposomal and free IL-2-Fc were comparable at 24 h. These new results are consistent with the findings from the confocal microscopy data at early times from Fig. 3, and are discussed in the revised text on p. 8.

4. The experiments presented in Supplementary Fig. 7 are excellent controls for understanding the role of each molecule in toxicity and the role of the liposome. It may be useful to present this data in the main figures.

We thank the referee for this suggestion. We moved original Supplementary Fig. 7 to the main text as revised Fig. 6 (Previous Fig. 6 was changed to Fig. 7).

5. The authors rely on the EPR effect for localization of the nanoparticles to the tumors. As they discuss in the final paragraph of the discussion, the EPR effect may be heterogeneous or not present in various tumors and patients. If the rationale is that systemic localization of the immuno liposomes provides the ability to target metastases effectively and reduce off-target toxicities, it may be necessary to discuss the challenges of translating therapies to the clinic that rely on the EPR effect more in-depth. It is also likely that the EPR effect plays even less of a role in small metastatic lesions and therefore the therapy may not be useful for micro-metastases. The role of the EPR effect in these small lesions should be discussed.

While it continues to be debated how homogeneous and effective the EPR effect is in human patients, there are several points that make us encouraged that the data shown here in mice could predict efficacy in humans. First is the key distinction between the drug agents/cellular targets of traditional nanomedicine— where typically chemotherapy or targeted drugs must reach tumor cells, and must deliver a functional dose of drug to every single tumor cell— vs. immunotherapy, where small amounts of potent cytokines/antibodies reaching a proportion of immune cells in the tumor microenvironment may have efficacy, or which may bind immune cells in circulation that carry these agents into the tumor. Notably, our data in a lung metastatic melanoma model (Figure 5) demonstrated that immunoliposomes could elicit superior therapeutic efficacy and survival benefits for treating metastatic lesions comparing to their soluble counterparts, in both small metastatic lesions (Figure 5a-e) and established metastatic nodules (Figure. 5f). Second, imaging-based clinical methods (e.g., MRI imaging of iron oxide nanoparticles to

identify patients exhibiting nanoparticle accumulation in tumors prior to nanoparticle therapy), are beginning to be employed clinically (Miller et al. *Sci. Transl. Med.*, 7(314), 314ra183–314ra183 (2015); Spence et al. *J. Control. Rel.*, 219(C), 295–312 (2015)), which pave the way for rational selection of patients for EPR-dependent therapies. Lastly, a variety of clinically-relevant strategies to increase the EPR effect in tumors are beginning to be defined, such as radiation therapy (Davies et al. *Cancer Research*, 64(2), 547–553 (2004); Lammers et al. *J. Control. Rel.* 117(3), 333–341 (2007)), photodynamic therapy (Kobayashi, H., & Choyke, P. L. *Nanoscale*, 8(25), 12504–12509 (2016)), and use of vasodilators (Tahara et al. *MedChemComm*, 8, 415–421 (2017)), to name a few. We have edited the discussion on pp. 15-16 to highlight these points.

6. The fate of the nanoparticles in the tumor is unclear. It would be helpful to discuss how exactly they interact with the target cells. Are they internalized by cells? If so, how do the liposome-anchored molecules interact with the target cells in the tumor? How long is the particle active in the tumor?

We have previously shown (Kwong et al. *Cancer Res.* 2013; Zheng et al. *J. Control. Rel.* 2013) that both IL-2-Fc-liposomes and anti-CD137-liposomes are relatively rapidly internalized on binding to lymphocytes. Thusfar we have found it challenging to precisely quantify their duration of activity in tumours. One approach that appears promising is monitoring downstream signaling from the IL-2 and CD137 receptors. We measured STAT5 activation (phosphorylated STAT5) in intratumoral T cells and NK cells following treatment with soluble or liposomal IL-2-Fc/anti-CD137, and found that pSTAT5 was still activated 24 hr after the last dosing in both groups, suggesting that stimulation is continued for at least 24 hr with both the soluble and nanoparticle immunotherapies (new Figs. 7j and 7k, p. 12.).

7. It is unclear whether statistical analyses were performed for each individual figure panel (e.g. 1g, 2 b, c, h, & 6d, h, i)

We have added the statistical significance for each individual figure panel.

Reviewer #2 (Remarks to the Author):

This manuscript reports that using liposomes to deliver surface-anchored IL-2 and anti-CD137 proteins reduces the toxic side effects observed when these agents are delivered as soluble factors. The evaluation of the immunoliposomes includes multiple tumor models as well as evaluation of biodistribution and effect on TIL and resulting cytokine production. The potential significance of the work is high and the quality is generally quite good.

We thank the reviewer for these positive remarks.

However, there are a few issues that should be addressed before publication:

1. It is really unclear when immunoliposomes with fluorescently labeled antibodies were used, and in the experiments where they weren't used, how antibody loading was determined.

Fluorescently labeled antibodies were used in the PK studies (Fig. 2d-e), biodistribution studies (Fig. 2f-i), confocal tissue imaging (Fig. 3) and some flow cytometry analysis (Fig. 2j). In the experiment where they were not fluorescently labeled, the antibody loading was determined by an infrared spectroscopy protein assay, as mentioned in the methods– “The quantity of antibodies conjugated to the liposome surfaces were determined by a Direct Detect infrared spectrometer (EMD Millipore)”. These details have been clarified further in the Methods.

2. Biodistribution was determined by fluorescent labeling of immunoliposomes followed by whole organ fluorescence imaging using a Xenogen imager. The accuracy of this method should be demonstrated and verified, since it is non-traditional compared to direct radiolabeling and tissue quantification.

Biodistributions were determined by fluorescent labeling of immunoliposomes followed by whole organ fluorescence imaging using infrared dye-labeled proteins and a LI-COR Odyssey Imaging System, not a Xenogen imager. IRDye CW800 was used as a fluorophore to label anti-CD137 or IL-2-Fc. It is an infrared dye with excitation wavelength at 800 nm, a wavelength with minimal autofluorescence and absorption by biomolecules in tissues. The amount of accumulated antibodies in each organ was calculated from a pre-determined standard curve for each organ. Standard curve were calculated using known doses of IRDye CW800 infrared dye injected into different organs and the fluorescence intensities were measured by the LI-COR Odyssey imaging system. We have now added these standard curves for organs and PBS as **new Supplementary Figure 8**. The linearity of the standard curves is all good at a wide range, suggesting this is an accurate way to measure the infrared dye in tissues. These details have also been clarified in the Methods **on p. 21**.

3. Importantly, no dosing studies are reported. It is very possible that the toxic side effects that are observed with the free proteins versus the immunoliposomes is simply because the free proteins were administered at too high a dose. In drug delivery, liposome delivery usually results in reduced side effects but also reduced efficacy; therefore investigators will often dose at a high dose so that efficacy is observed with their liposome formulation but toxicity is observed with the free drug. This is artificially biases the outcome to be favorable to the liposome formulation.

We actually did perform limited dose response studies, for the exact reason cited by the referee, shown in Supplementary Fig. 1 (discussed in the text on p. 4). We found that reducing the dose of soluble IL-2-Fc and anti-CD137 could reduce the systemic cytokine storm and weight loss induced by the free drugs, but this reduced toxicity was accompanied by parallel loss in efficacy, such that a lower dose (1/3 the dose used for the bulk of the studies) was not toxic but also not nearly as effective as the higher liposomal dose of the two agents. Likewise, altering the dosing schedule did not allow the soluble drugs to be safely administered. We thus conclude that the liposomes outperform the free drugs, whatever the dose selected for the free cytokine/antibody.

4. The extension to humans is not straightforward since the EPR effect can be strong in mouse models given the right tumor model, but is usually not observed as robustly in humans. This should be mentioned, since the advantages presented here depend on a strong EPR effect.

Please see the response to reviewer #1 point 5.

REVIEWERS' COMMENTS:

Reviewer #2 (Remarks to the Author):

the authors have adequately addressed my comments and I recommend publication.

RESPONSE TO REVIEWS

Reviewer #2 (Remarks to the Author):

the authors have adequately addressed my comments and I recommend publication.

We thank the referee for the positive comment.